# Improving Polysaccharide-Based Chitin/Chitosan-Aerogel Materials by Learning from Genetics and Molecular Biology

**DOI:** 10.3390/ma15031041

**Published:** 2022-01-28

**Authors:** Matthias Behr, Kathirvel Ganesan

**Affiliations:** 1Institute of Biology, Leipzig University, Philipp-Rosenthal-Str. 55, 04103 Leipzig, Germany; 2German Aerospace Center, Institute of Materials Research, Linder Höhe, 51147 Cologne, Germany; k.ganesan@dlr.de

**Keywords:** wound healing, materials, skin substitutes, aerogel, chitin, chitosan, insect, drosophila, proteins, obst-A

## Abstract

Improved wound healing of burnt skin and skin lesions, as well as medical implants and replacement products, requires the support of synthetical matrices. Yet, producing synthetic biocompatible matrices that exhibit specialized flexibility, stability, and biodegradability is challenging. Synthetic chitin/chitosan matrices may provide the desired advantages for producing specialized grafts but must be modified to improve their properties. Synthetic chitin/chitosan hydrogel and aerogel techniques provide the advantages for improvement with a bioinspired view adapted from the natural molecular toolbox. To this end, animal genetics provide deep knowledge into which molecular key factors decisively influence the properties of natural chitin matrices. The genetically identified proteins and enzymes control chitin matrix assembly, architecture, and degradation. Combining synthetic chitin matrices with critical biological factors may point to the future direction with engineering materials of specific properties for biomedical applications such as burned skin or skin blistering and extensive lesions due to genetic diseases.

## 1. Introduction

Biopolymers are polymers of natural origin and can be found in wood, plants and animals. Chitin is the second-most abundant, non-toxic natural biopolymer containing N-acetylglucosamines (GlcNAc) linked by β-1,4 glycosidic bonds from which chitosan can be derived by N-deacetylation [1,2]. Polysaccharide-based chitin/chitosan materials are increasingly used for future human-made products [3,4]. The reasons for this are manifold, since chitin/chitosan matrices can be flexible and, at the same time, very robust and scratch-resistant [5,6]. In addition, they exhibit different properties in their permeability to molecules [7]. Furthermore, they offer incredible possibilities from transparency to multiple color arrangements [5,6]. Finally, chitosan possesses antimicrobial, antioxidant, anticarcinogenic, and anti-inflammatory properties [8,9].

Natural chitin polymer chains are helices in which each sugar unit is inverted with respect to its neighbors. This leads to high stability as the rigid ribbons are connected to hydrogen bonds [10]. Linear chitin polysaccharides occur naturally in three crystalline allomorphs known as *α*-, *β*-, and *γ*-chitin. They contain different orientations of the microfibrils [2,11,12,13,14,15,16,17,18,19]. The chitin sheets arrange in layers that show antiparallel orientation in *α*-, and *γ*-chitin and parallel orientation in *β*-chitin [20]. This arrangement has consequences on chitin properties. The *α*-chitin has the strongest inter-sheet and intra-sheet hydrogen bonding. In contrast, the β-chitin has weak hydrogen bonding by intra-sheets and is characterized by a weak intermolecular force [21,22]. Chitin/chitosan can be extracted from various natural sources, such as crabs, lobsters, shellfish, algae, and fungi. However, access to a high amount of natural chitin/chitosan is limited since the most abundant sources are part of a global sea food market [23,24,25,26,27,28,29].

Insects are a terrestrial source since their cuticles and wings contain chitin/chitosan [30]. However, this requires the cultivation of a high number of adult or pupal animals. Another alternative source could be microbes. The cell walls of fungi contain pools of chitin/chitosan, but chitin/chitosan production is not yet scaled up to an industrial level [31]. However, chemical extraction processes from such natural sources involve using aggressive chemicals at high temperatures for extended periods, which can alter the properties of chitin/chitosan [32]. Recent studies show a more ecofriendly extraction of chitin/chitosan from natural sources by fermentation using lactic acid bacteria [33]. Their main chitosan/chitin degradation products are oligosaccharide/chitooligosaccharide (COS), which also possess remarkable biological and biomedical properties. COS has a low molecular weight (Mw), high degree of deacetylation (DD), high degree of polymerization (DP), low viscosity, and complete water solubility [34]. The genetic engineering of bacteria plays an essential role in chitin bioconversion to COS products, servings as a primer for synthesizing complex carbohydrates [35].

Given the different options for obtaining chitin/chitosan or their degradation products, the challenge remains to produce chitin/chitosan-derived matrix materials for target-specific biomedical usage. The synthetic production of chitin/chitosan matrices could be one elegant alternative (Section 5). To further improve the properties of these synthesized matrices for biomedical applications (Section 2 and Section 3), the addition of bioactive molecules and proteins that control the matrix architecture provides an exciting option (Section 4, Section 5, Section 6 and Section 7).

## 2. Chitin/Chitosan Matrices Provide the Potential for Multiple Biomedical Applications

Due to their high biocompatibility, chitin and chitosan have shown great potential for various uses in the medical field [36]. The positive charge of a chitin/chitosan-based matrix due to the protonation of amino groups is an essential property [37]. This increases the solubility of chitosan and usually occurs in acidic environments [38]. Because of the positive charge, chitosan can form complexes with numerous negatively charged molecules, such as growth factors, nucleic acids, and cytokines [39,40,41]. Thus, chitin/chitosan matrices recruit bioactive factors from the environment, protect their degradation and increase local efficacy [36]. In addition, protonated chitosan can interact with extracellular matrix components such as glycosaminoglycans and proteoglycans [38,41]. Furthermore, the possibility of enzymatic degradation of chitosan plays a decisive role. Lysozyme (muramidase) is present in mucous membranes of the human intestine and other mucosal epithelial cells. This enzyme separates the bonds between acetylated units and thus cleaves chitin/chitosan into oligosaccharides [42,43,44,45,46,47]. Moreover, combining chitin/chitosan matrices with other polymer materials such as gelatin or collagen may improve their mechanical and biological properties [7,48,49,50]. Finally, synthetically engineered chitin/chitosan-produced materials show biocompatible, biodegradable, and non-toxic properties.

Biopolymers show multifunctionality as beads, microparticles, nanoparticles, micelles, hydrogels, tablets, and capsules. Chitin-based biopolymers can be formed into nanogels, namely hydrogels confined to the nanoscopic range. Chitin nanogels possess valuable attributes, such as the ability to load drugs, a large surface area for bio-conjunction, size tunability, controlled release, and an excellent ability to respond to environmental stimuli [51,52,53]. Chitosan gels show bioavailability for drugs, enzymes, and proteins [54]. In summary, films, beads, fibers, intragastric floating tablets, microspheres, and chitin/chitosan nanoparticles and their derivatives have been formulated for use in the pharmaceutical field. Recent findings even support their potential for gene therapy applications [7,40,55,56,57,58,59,60,61,62,63].

Studies over the past decade showed that the biomedical advantages of synthetic chitin/chitosan matrices are dependent on cell type, location and time of treatment, and limitations of the matrix itself [64]. The molecular weight (MW), which is affected by proteinization and degree of deacetylation (DD), can influence the properties of the chitin/chitosan matrix, e.g., solubility, viscosity, and biological activity [64]. The potential for gene delivery by chitosan/siRNA nanoparticles was supported by high MW and DD in human lung carcinoma cells [65]. In contrast, lower MW and DD showed a better hypocholesterolemic effect than high MW and DD in rats [66]. Moreover, these and other data, including the possibility of binding growth factors required for wound healing and bone regeneration, indicate the potential of chitosan-based grafts and nanoparticles for delivery of drugs and oligonucleotides (siRNA, etc.) for other biomedical applications [52,64,67,68,69,70].

The subject of current research is the improvement of synthetic chitin/chitosan matrices for various biomedical applications: for instance, chitosan displays biocompatibility during dental pulp regeneration of immature dog teeth [71]; chitin-nanocomposites showed improved blood clotting ability [72]; chitosan-gels induce angiogenesis in rabbit models [73]; application of chitosan hydrogels accelerates dermal wound healing in rats [74]; and chitosan–fiber scaffolds support attachment and differentiation of human monocytes, which is a central requirement for successful bone tissue engineering [75]. Chitosan-blended bacterial cellulose (cellulose synthesized by *Acetobacter xylinum*) composite films provide potential applications, such as wound dressing, arterial vascular grafting, and delivery systems of drugs and proteins [76]. Moreover, depending on its molecular weight, chitosan shows proinflammatory properties affecting the migration of leukocytes and macrophages, which is essential for wound healing and tissue regeneration [77,78,79,80]. However, proinflammatory properties could cause allergic reactions and restrict biomedical use [64]. For example, chitosan (Heppe Medical Chitosan) affects the integrity of the nasal epithelium by reducing transepithelial electrical resistance through the degradation of the tight junction protein occludin. This intranasal application facilitates transepithelial allergen passage in the mouse model [81]. In contrast, well-known allergies against seafood crustaceans are likely caused by the muscle protein tropomyosin and, are therefore, unrelated to their chitin/chitosan [82]. Moreover, crab shell has been clinically used for many years as natural chitin matrix wound dressing graft material in Japan, without reported allergic incidents [64]. Furthermore, chitin films showed no allergenicity upon dressing wounds in rats [83]. There is no apparent effect of chitosan in terms of allergy or severe inflammation [9]. Collectively, chitin/chitosan matrices and composites are promising materials for active pharmaceutical and other bioactive agents needed for orthopedic and periodontal applications as well as for tissue engineering and wound healing [53,55].

## 3. Natural and Synthetic Chitin/Chitosan Matrices Support Wound Closure

A closer look at the biological function reveals interesting analogies between the multilayered epidermis of vertebrate skin and layers of the epidermal epithelium, with its overlaying lamellar chitin matrix, in insects [84]. The chitin matrix forms the central barrier at the insect cuticles [85], similar to the suprabasal cell layers of the human epidermis [84,86]. Loss of the chitin matrix and defective skin epidermis compromises the critical barrier function against dehydration and harmful environments. This can have devastating consequences for animals and humans. Atopic dermatitis is a broad term for many defects of the epidermis, such as epidermolysis bullosa simplex (EBS) or junctional epidermolysis bullosa (JEB). Patients suffer from chronically swollen and cracked skin areas with fragile barriers and devastating inflammations [86,87]. Healing of such large areas of skin is often not possible, taking months or longer, and requires enormous medical efforts in rare genetic diseases, including even gene therapy interventions. [87].

Chitin and chitosan have been shown to be useful as wound dressing materials. For example, the used inner chitinous membrane of soft crab shell carapaces showed advantages in wound healing [88]. Cells show the capacity to spread, proliferate, elongate and adhere to the used chitinous membrane, and in the rabbit model, it promoted faster epidermal wound closure. Enhanced cell adhesion could be the underlying molecular reason due to specific matrix surface structures. In addition, increased numbers of neutrophils and macrophage immune cells infiltrate the epidermis, and after two weeks, the chitinous membrane fell off without showing any side effects in the rabbit model [88]. In the mouse model, chitosan dressing of grafts directly promoted skin wound healing by affecting growth factor expression [89]. Macroscopic and histological analysis demonstrated the role of chitosan hydrogel in the treatment of dermal burns through the induction of full-thickness transcutaneous dermal wounds in Wistar rats [74]. Moreover, chitin gauze and spray revealed satisfactory healing of various traumatic wounds in humans [90]. Chitin/chitosan in artificial grafts taping such wounds could be used to re-establish skin architecture [74]. Adding bioactive molecules, such as antimicrobial agents and growth factors promotes skin cell growth, regeneration, and anti-inflammatory effects [91]. Thus, clinical models show that gels based on chitin nanofibrils accelerate the healing of wounds in humans. Their biochemical activities lead to activation of polymorphonuclear cells, fibroblast activation, cytokine production, migration of giant cells, and stimulation of type IV collagen synthesis. In addition, it prevents hypertrophic scarring and keloid scars [92]. Techniques of 3D printing, electrospinning, and aerogels with chitin/chitosan will improve the properties of wound materials [7,50]. Production of a very powerful chitin/chitosan matrix could be even suitable in cases where it could act as a long-lasting graft-like tissue forcing wound closure while also protecting as a barrier.

Skin grafting is essential for reconstructive surgery in patients with burns, trauma, and non-healing or large wounds. Wound healing comprises many aspects, namely cleansing, granulation/vascularization, and re-epithelialization. These depend on an optimal microenvironment and cytotoxic factors [93,94]. However, the main phases of wound healing, namely inflammation, new tissue formation, and remodeling, are all affected by mechanical forces. Indeed, mechanotransduction is a potential target of excessive scar formation, a significant clinical and financial burden, and requires improved therapies to reduce scarring [95]. Thus, mechanical offloading can provide access to minimize scar formation and advance adult wound healing. To meet the specific and individual medical requirements, the production of a synthetic chitin matrix with defined mechanical properties will be a future challenge. Therefore, we focus on a concept combining chemical and molecular biology tools to develop a new perspective for generating improved materials.

## 4. Using the Natural Toolbox to Improve the Quality of Synthetic Chitin/Chitosan Matrices

Polysaccharide-based aerogels are porous materials with a net-like nanostructure. Aerogels exhibits high porosity, low density, thermal conductivity and large surface area. In addition, they are ultra-lightweight, have high water absorption capacity, excellent shock-absorbing properties, and are flame and moisture resistant. Overall, these multiple special properties make them suitable for numerous applications in the medical, pharmaceutical, and cosmetic sectors, among others [4,96]. The promising use of natural and synthetic aerogels for fluid management, healing, and regeneration of wounds led to the increasing trend of articles published in recent years [4,96,97,98]. Many promising examples provide evidence for the multiple usage of synthetic chitin/chitosan-based matrices, such as chitosan hydrogels, films, porous scaffolds, textile fibers, and others, and the capacity to optimize their functionality and bioactivity as biomedical materials [73,74,75,99,100,101]. The numerous synthetic chitosan-based scaffolds used to promote wound healing were recently summarized [7]. Chitin/chitosan-aerogels show advanced properties, such as inhibiting the *Staphylococcus aureus* growth, a widespread cutaneous pathogen responsible for the great majority of bacterial skin infections in humans [102,103]. Moreover, the aerogel technique provides powerful tools for technical modifications, such as blending, crosslinking, and modifying microstructural and physical properties to produce chitin/chitosan matrices (see Section 5).

Hou and colleagues state in their recent opinion that for the future’s advanced technologies, chitin will ultimately drive many innovations and alternatives using biomimicry in materials science [6]. Indeed, natural chitin/chitosan matrices form incredibly different properties, depending on the location and developmental timing at which animals need them [104,105]. For example, natural chitin/chitosan matrices form the outermost body barrier or cell wall, protecting organisms, shaping their organs, or even serving as tight claws or teeth [106,107]. They even cover soft wings or form the transparent layer of compound eyes of insects [108]. Moreover, thinking about the beauty of butterflies, chitin offers a unique color palette that is durable and responsive. The coherent scattering in the periodic arrays of multilayered chitin/chitosan matrix structures creates countless color combinations [6]. Recent investigations of different biological disciplines concluded that the high performance of natural chitin matrices relies on their specialized structures, the perfect packaging of the chitin/chitosan chains into long fibers that must assemble into complex architectures [109]. A key role lies in the molecular mechanisms, namely the genes coding for the key proteins and enzymes that establish natural chitin matrices, mature them, and degrade them without preventing newly formed matrices from keeping epithelial barriers [110,111,112].

In our view, the numerous variations of chitin matrix architecture in insects suggest a bioinspired view of the natural molecular toolbox (see Section 6). Synthetic chitin matrix scaffolds such as aerogels can be improved and adapted to the specific medical application of wound healing by adding proteins that determine the matrix architecture and thus reducing tensile stress, improving wetting and impermeability to pathogens and toxins. The improved materials could then enable more efficient wound healing with as little scarring as possible.

## 5. Production of Polysaccharide-Based Chitin/Chitosan Aerogel Matrices

### 5.1. Engineering Synthetic Chitin/Chitosan Matrices

The optimization of synthetic matrices for biomedical applications is a demanding challenge. Nature’s highly hierarchical architecture of chitin matrices could be a significant obstacle for engineering synthetic materials. However, generating a synthetic material based on the inspiration of biological chitin/chitosan matrices with structural and functional properties is challenging. Therefore, as a prerequisite for developing future bioinspired materials with synergistic properties of the chitin/chitosan peptide matrix system, various materials still need to be explored. Various methods of wet-gel matrix formations are listed in Table 1.

Engineering randomly connected fiber matrices of chitin/chitosan into various dimensions can be achieved in laboratories by inducing either the self-assembly of the molecular chains or percolated network formation of nanowhiskers/nanofibrils. Van der Waals physical bonding or covalent binding between molecular chains is significant for forming matrices. There is a series of actions happening in the wet state-forming higher-order structures of chitin/chitosan matrices: (a) association of molecular chains in a fashion producing the smallest building blocks called nanofibers; (b) formation of a randomly interconnected three-dimensional network in a small domain; (c) finally the complete formation of pure matrices of chitin/chitosan filled with liquid. The following parameters are the actors controlling molecular chains from their solute state to the fibrillar network: temperature, pH, chemical crosslinking, and a non-solvent approach [113,114,115].

### 5.2. Blending, Physical and Chemical Crosslinking of Chitin/Chitosan Matrices

Recent reports show modifications of chitin/chitosan matrices with desired peptide/protein molecules [116,117]. Chitin/chitosan-bound peptide/protein matrices were developed by blending the molecular chains or inducing the physical (Van der Waals forces) or chemical crosslinking between the molecules. Compared with chitin, chitosan can be used for synthesizing bound chitosan-peptide/protein matrices because of abundant deacetylated amine functional groups. The amine functional groups of chitosan can be activated for blending or physical crosslinking by protonation. Physical crosslinking, such as through Van der Waals force interaction, occurs between chitin/chitosan and protein, forming polyelectrolyte complexes [72,118,119,120,121,122]. Cost-effective methods have been reported to produce polyelectrolyte complexes as dried membranes [119] or scaffold materials (by freeze-drying) [118] by blending chitosan with silk fibroin. In this case, dried membranes and scaffold materials of blended polyelectrolytes have not displayed nanofibrillated gel networks. However, the blended materials have exhibited efficient wound healing properties [119], enhanced biological activity, and suitability as supporting materials for tissue engineering [120].

Compared with physically bound gel matrices, chemical binding between chitosan and peptide/protein provides high stability and enhanced mechanical properties. In general, the chemical crosslinking reactions are carried out, forming covalent bonds between chitosan and peptide/protein molecule by either coupling with an anchoring spacer group/crosslinker between them or eliminating the water molecules by amidation [123,124,125,126,127,128,129]. In this case, a robust three-dimensional hydrogel network can be achieved, mimicking the extracellular matrix, promoting the adhesive property of cells and biological functions. As the crosslinkers do not play any vital role in the desired biological function, the crosslinker should be a non-toxic reagent. In recent reports, a nature-based crosslinking agent, genipin, was used to produce chitosan–protein hydrogel [124,130,131]. In the case of chitosan–collagen–hyaluronic acid, after increasing the crosslinking concentration by genipin, the cell spreading capacity was improved although the water uptake capacity of hydrogels was reduced [124].

No crosslinkers are necessary when chitosan can be directly bound with peptide/protein by amidation [126,132]. The most employed reagents for the amidation reaction are N-hydroxysuccinimide or derivatives of it (activating the amine groups of chitosan and carboxylate groups of peptide/proteins) and water-soluble carbodiimides (scavenging the water molecules turned to be urea derivatives). The covalent crosslinking of carboxymethyl chitosan with collagen peptides using water-soluble carbodiimide (1-(3-dimethylaminopropyl)-3-ethylcarbodiimide hydrochloride) and N-hyydroxysuccinimide was recently shown [132]. Further purification of hydrogels is essential to use the matrices for biological applications. The workflow and methods for preparing chitin/chitosan-aerogels are summarized in Figure 1

### 5.3. Modifying Microstructural and Physical Properties of Synthetic Matrices

The morphology and topography of the chitin/chitosan matrices receive close attention in designing the wound dressing materials. When ambient drying or vacuum-drying the wet-gel matrices of chitin/chitosan materials, the existing surface tension between solid, liquid, and gas can pull the nanostructures together. At the same time, the fluid becomes a gas, which destroys the natural structure [134]. Therefore, it results in more than 90% of volume shrinkage. These materials are named xerogels [135]. The complete collapse of pores can be confirmed by nitrogen adsorption-desorption analyses, mostly showing no considerable values if the solvent medium in the matrix is water or ethanol [135,136]. The methods for preserving pores employed in the laboratories are freeze-drying (sublimation of frozen water to gas) and supercritical drying (solvent exchange of solvent with supercritical gas [134,136]. They fabricate highly porous materials as the fluid in the wet-gel nanostructures of chitin/chitosan can be replaced with air [54].

In the freeze-drying technique, the natural gel nanostructures are destroyed due to Vander Waals interactions, promoting significant aggregation of nanofibers. As a result, the specific surface area becomes low (<110 m^2^/g) [136]. However, due to the frozen water molecules (ice crystals), the pores generated by sublimation can display the material as being highly porous. These foam-like materials are named cryogels [134,136]. In some reports, it was demonstrated that use of tert-butyl alcohol instead of water in the gel matrix could prevent structural damage [137]. Due to the low surface tension and high freezing point of tert-butyl alcohol compared to water, freeze-drying reduces the capillary forces between the solvent and the gel matrix and maintains the three-dimensional network structure [137].

Supercritical drying can be employed in which the fluid in the gel matrix is converted to gas in its supercritical state [134]. No existence of surface tension favors preserving the microstructure of the gel matrices. In supercritical drying, an environment-friendly supercritical gas such as carbon dioxide is employed. In this drying process, the fluid in the gel matrix is switched to a suitable solvent such as simple alcohol (e.g., ethanol), which has good miscibility with CO_2_. Then the alcohol in the gel matrix is saturated with CO_2_ under supercritical conditions (>31 °C and >73.8 bar) in an autoclave [134].

Finally, supercritical carbon dioxide gas is degassed at above 40 °C, and the matrix is filled with air [134]. Approximately 5–40% of volume shrinkage can be observed in this process depending upon the matrix’s mechanical strength and physical properties. For instance, examples of aerogel of chitosan derivate which was dried under supercritical conditions, are shown in Figure 2A [133]. As a result, an open porous structure of the chitin/chitosan matrix can be achieved with properties close to the reality in the hydrogel or wet-gel media [54].

There are huge morphological differences between ambient dried, freeze-dried, and supercritically dried materials [136]. Xerogels, cryogels, and aerogels do not show similar interactive properties with the wound surface. For instance, Guo et al. demonstrated in the recent report that the morphology of aerogels of chitin promoted the healing process, accelerated macrophage migration, enhanced fibroblast proliferation, induced collagen deposition, and promoted granulation and vascularization in comparison with cryogels. Moreover, after 11 days of the wound healing process, the scar area appeared reduced and was significantly smaller in the case of aerogels than cryogels. [138].

In the last decades, the dry membrane, scaffolds, fibers, and hydrogels of chitin/chitosan in wound healing applications have been explored, and their mechanisms have been analyzed in vitro and in vivo [55,80,139,140,141,142]. Nevertheless, the finely distributed nanostructures and open porous network of aerogels of chitin and chitosan will be a new class of porous materials for this field of application (examples are listed in Table 2). The aerogels of chitin/chitosan functionalized with protein/peptides are promising candidates for wound healing applications. The materials possess specific surface area, high sorption capacity, biocompatibility, antimicrobial property, interactive surface functionalities, and a structural similarity to the skin’s extracellular matrix [4,138].

## 6. The Molecular Toolbox for Natural Chitin Matrix Production

The already nature-like aerogel matrix provides an optimal synthetic basis for implementing a bioinspired version. To go even further, the biological knowledge of how natural matrices can take on different structures and their associated properties will be very useful. Chitin-based matrices are the most prominent part of the arthropod cuticles [147,148]. The chitin matrix has two main functions: it forms an exoskeleton that shapes the organs/animals, and it protects the organs from various stresses, such as mechanical impact, invading pathogens, toxins, and desiccation at high temperatures [84,149,150,151]. Arthropod cuticles repeatedly molt during the transition of larval/nymph stages to fit their increasing body size, which is under systematic hormonal control [148,152,153,154,155,156].

### 6.1. The Architecture of Cuticular Chitin Matrices

In arthropod cuticles, including insects, antiparallel arranged *α*-chitin chains are most abundant [157]. About 18 to 25 chitin chains assemble into nanofibrils (~3 nm diameter; ~0.3 µm length) [158]. Chain assembly ultimately requires proteins to encase the nascent nanofibrils. At the next hierarchical level, the nanofibrils assemble into long chitin-protein fibers arranged in horizontal planes. Nanofiber stacks form a plywood structure, which is a prominent part of many cuticles [110,147,159]. The hierarchic arrangement, the stiffness, strength, fracture toughness, and the working fracture of insect chitin matrices have been discussed previously [110,160,161,162,163,164].

### 6.2. Natural Chitin Production

The common feature of epithelial cells is their apical-basal polarity, with the apical cell side facing outward (epidermis) or into the lumen of tubular organs. Located in the apical membrane, chitin synthases (CHS) synthesize the chitin chains [1,165]. The latter can form initial chitin fibrils in a self-assembly process [111]. Dynamic apical cell surface structures (microvilli) can determine the boundary conditions for chitin-fiber self-assembly [112]. However, the synthesis and initial self-assembly of chitin polymers are not enough to form chitin matrices with defined architecture and properties.

### 6.3. Proteins Control Proper Chitin Matrix Formation

Importantly, the volume fraction of the constituents of the chitin–protein fibers significantly influences the matrix performance [166]. For example, the epidermal chitinous-matrix is relatively stiff, very dense, and robust against diverse stresses [84,85,167]. This requires strong chitinization with a lamellar, multilayer chitin matrix arrangement (Figure 2B–D) [85,105,168]. In contrast, the chitin matrices of tubular organs are relatively soft and only single-layered [164]. Examples include the respiratory tracheal system and the digestive system [169,170,171,172,173,174,175,176]. The tracheal and foregut chitin matrix must seal tube lumina for specific transport processes while being flexible [177,178,179,180]. The peritrophic matrix surrounding the midgut food is a chitin matrix that needs to be selectively permeable for nutritional uptake [180]. In all cases, chitin chains are embedded in a protein matrix [147,159,180,181]. Those proteins control the formation of the above-described rigid/stiff and also flexible chitin matrices, and are therefore fundamental for controlling chitinous cuticle functions (Figure 2B,C) [180,182,183,184,185,186,187,188].

Insects possess a large number of cuticle proteins that contain chitin-binding domains and belong to different families [189,190,191,192,193]. Proteins with one or three chitin-binding domains (CBD) are members of the CPAP1 or CPAP3 family (Cuticle Proteins Analogous to Peritrophis) [194,195,196,197,198]. Those with a Rebers and Riddiford (R&R) chitin-binding sequence belong to the CPR family [192,193,195,196]. However, only a small number of the numerous chitin-binding proteins are critical for matrix formation (summarized in [111]). They mainly determine physiochemical properties such as the viscoelasticity and permeability of matrices. For example, loss of *Tribolium* CPR18 and CPR27 results in wrinkled elytra [199], and CPAP-1C, -H, -J knockdown showed fragile chitin matrix and structural integrity effects [194]. The CPAP-analogous *Drosophila* genes were identified as the *obstructor* (*obst*) gene family, consisting of 10 members, each with three chitin-binding domains [200]. Null mutant studies of three members, Obstructor (Obst)-A, Obst-C, and Obst-E, confirmed their vital and critical functions in assembling chitin matrices during insect development [198,201,202]. The mutant studies further prove that the chitin matrix assembly and packaging into higher-ordered structures are prerequisites for cuticle stability, tightness, protecting function, and insect survival [183,202].

At the molecular level, Obst-A binds chitin and provides a scaffold that embeds chitin chains/fibrils. In addition, Obst-A localizes the chitin deacetylases Serpentine (Serp) and Vermiform (Verm) within the scaffold, enabling optimal chitin/chitosan matrix maturation. Chitin deacetylases catalyze the removal of acetyl groups from chitinous substrates and play critical roles in shaping the chitin matrices of cuticles in *Drosophila* [174,175,203] and other insects [204,205,206,207,208]. To further package and protect the chitin matrix, Obst-a recruits Knickkopf [183,202]. Knickkopf (German term for buckling head) protects newly synthesized chitin fibers from chitinases and organizes fibers into thicker and longer bundles that can stack into horizontal lamellae [182,209,210]. The Obst-A, Serp, Verm, and Knickkopf proteins represent a molecular core complex unit that acts as a hub for chitin matrix assembly, packaging, and protection (Figure 3). This protein complex mainly acts at the apical cell surface in the so-called cuticle assembly zone. Accordingly, proteins and enzymes enrich apically at the cell surface within the cuticle assembly zone. Therefore, these hub proteins offer an exciting starting point for improving synthetic chitin matrices to achieve modifications and setups of highly complex structures with defined properties.

Chitinases (Chts) belong to the large glycosylhydrolase family 18, widespread in the animal kingdom [211,212]. Usually, they are known to degrade chitin matrices by hydrolyzing chitin chains into smaller oligosaccharides. However, it remained unclear why insects possess numerous chitinase genes. Genetic studies uncovered a set of core chitinases (Cht2, Cht5, Cht7, Cht12) that degrade tight lamellar and soft non-lamellar chitin matrices. They contain various features, including (multiple) catalytical domains, transmembrane or chitin-binding domains that guarantee substrate specificity [85,164,213]. In contrast, *Drosophila* Cht2 and *Tribolium* Cht7 are also required for chitin matrix assembly [164,214,215]. Both influence chitin chain assembly into higher structures by potentially limiting the increasing lengths of nascent chains and resulting fibers (Figure 3). Whether Cht2 and Cht7 belong to the hub of chitin matrix formation remains elusive.

## 7. Advantages and Limitations of Chitin/Chitosan-Protein Materials

The development of biomedical materials is challenging. These materials require surfaces that enhance cell attachment and maturation [73]. Therefore, using chitin-binding proteins and enzymes, such as deacetylases, that control the surface architecture of natural chitin matrices could be an attractive new option to improve synthetic chitin/chitosan-based aerogel materials. However, despite all the confidence in coupling proteins to produce synthetic chitin/chitosan–protein matrices, some technical limitations must be considered. First, proteins are more than just large peptides and long chains of amino acids. Proteins are polymers that must fold into and retain their native three-dimensional structure to be bioactive during and, if desired, after production. Thus, and second, synthetic production or in vitro expression systems must allow the isolation of natively folded proteins. Third, the expression systems must modify proteins post-translationally, e.g., to carry out phosphorylation and glycosylation. These properties are achievable using the well-established insect cell expression systems and the baculoviral technique [216]. Fourth, exposure to harsh solvents must be avoided during engineering steps, as they could damage protein bioactivity. Fifth, the optimized ratio of proteins to each other and the chitin/chitosan molecules is uncertain and needs to be clarified.

When coupling high molecular weight protein with chitin/chitosan nanofibers, retaining protein physical and structural properties is critical for biological applications. It is possible to surpass it with small proteins or peptide molecules (oligo-/polypeptides), which enhance the interaction of biological molecules with artificial chitin/chitosan matrices. However, most cuticle proteins are not high-molecular-mass proteins (e.g., MW > 100 kDa) but are much smaller, as is the case for Obst-A (237 amino acids; 27 kDa), and are therefore likely to be easier to handle. Alternatively, instead of coupling the full-length Obst-A protein, its critical chitin-binding domains, which are evolutionarily well-conserved, could be used. Obst-A contains three short chitin-binding domains (57/59/67 amino acids), each of which binds in vivo and in vitro to colloidal chitin (unpublished results MB). The coupling of these domains, either as single or multiple domains, could alter the surface properties of porous chitin/chitosan materials. Concerning the material’s cleanliness, quality, physical and structural properties, the synthesis of artificial material will be limited to the aqueous medium.

The initial interactions are determined by the surface energy, chemical composition, stiffness, structure, and topography of the biomaterial surface in contact with the biological environment [73]. Hydrophobic or charged surfaces of the materials attract bioactive molecules. Such bioactive molecules are human extracellular matrix proteins, growth factors, short bioactive peptides, and synthetic chemicals, which specify the materials for the requested biomedical application. The addition of these bioactive molecules could further improve chitin/chitosan materials [73]. Additionally, it is possible to combine these techniques, but this requires many steps, which could be disadvantageous.

Importantly, aerogels are excellent materials to entrap, reinforce, or even interact with proteins, drugs, or bioactive material, making them a fascinating biomedical product [4,138]. Therefore, the potential of the interaction of chitin controlling proteins that modulate aerogels is one essential thesis of our final conceptual model (Figure 4).

## 8. Conclusions

The skin is our largest organ and is vital for our survival. However, wound healing of large areas of skin is very problematic, takes very long periods, and requires enormous efforts, or even gene therapy interventions in rare genetic diseases. With the increasing knowledge of chitin/chitosan science regarding structural and functional designs for biological application, customized chitin/chitosan matrices could be generated. The molecular control of the architecture of chitin matrices is a key to specialized functions [110,217]. In insects, a small set of chitin matrix core complex unit proteins controls chitin matrix properties precisely [85,164,182,183,186,204,205,209,215,218]. Thus, we are pursuing a strategy of tailoring synthetic chitin/chitosan molecules at the molecular level with peptide/protein molecules that may be transferred to stable matrices through molecular assembly processes and the formation of nanofibrillar structures. Exploring individual matrix concepts such as bioinspired chitin/protein matrices will show us new ways to shift the paradigm from purely curative medicine to preventive medicine, from a one-size-fits-all approach to personalized therapies (Figure 4).

## Figures and Tables

**Figure 1 materials-15-01041-f001:**
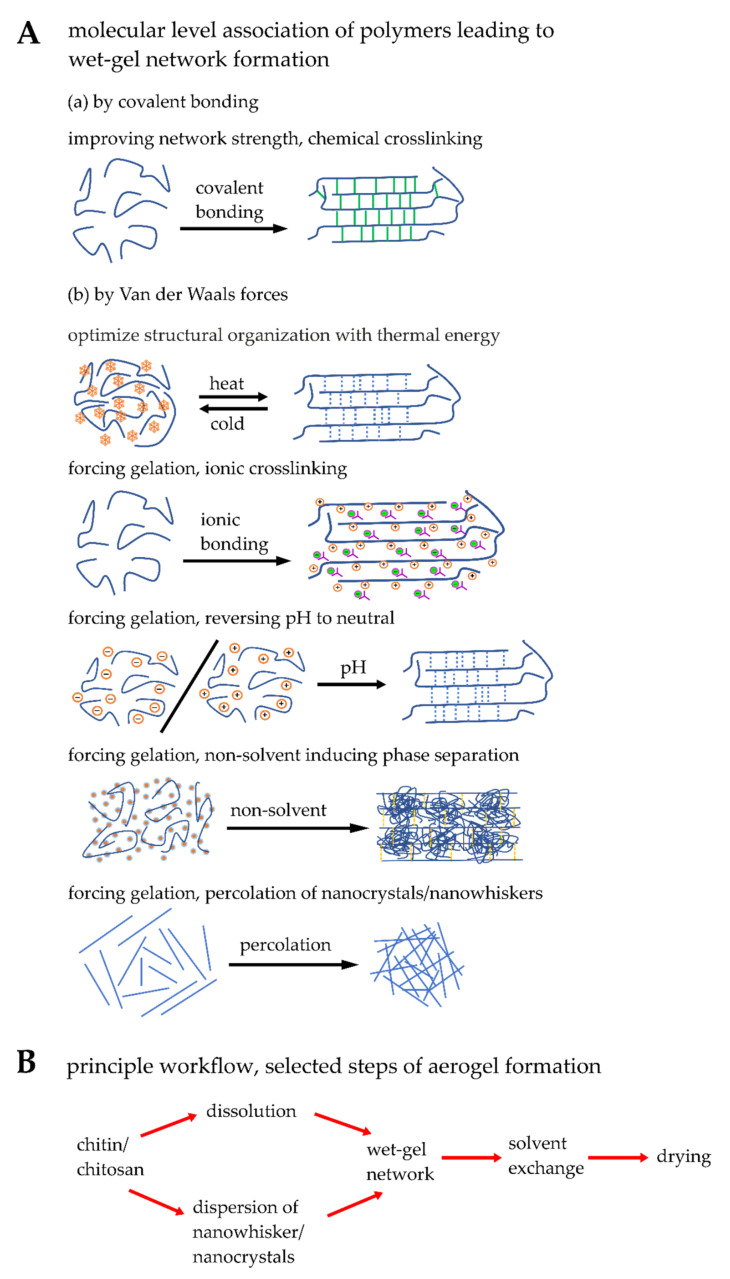
Schematic drawing presenting selected steps of a workflow of chitin/chitosan-aerogel production. (**A**) Network formation methods at the molecular level either by covalent bonding or by Van der Waals forces, mainly hydrogen bonding. (**B**) A general principle workflow of aerogel synthesis includes several steps: network-formation, solvent exchange, and drying. For more details, see [54,114,133].

**Figure 2 materials-15-01041-f002:**
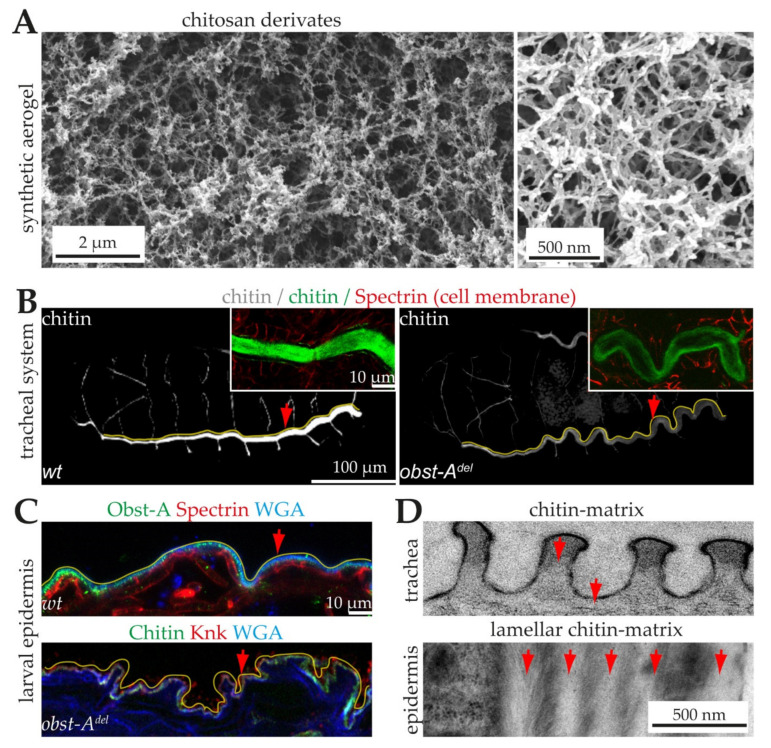
Synthetic and genetically controlled chitin matrices. (**A**) Scanning electron microscopy images of aerogel chitosan derivates [133]. Scale bars indicate 2 µm (left) and 0.5 µm (right). (**B**) 3D-projections of confocal Z-stacks of whole-mount immunostainings showing chitin (green) in late *Drosophila* embryos. Anterior is left, dorsal is up. An Alexa488-linked chitin-binding probe (green) detects chitin in the tracheal tube lumen. The main tracheal tube (marked by yellow lines) is straight and stable (arrow, left image) in wt but over-elongated and unstable with sinusoidal buckling (arrow, right image) in *obstructor (obst)-A* null mutants. Inlay, wt embryo shows a bright chitin staining (green) in the tracheal tubes, while the *obst-A* null mutant embryo contains only weak chitin staining (green), indicating unusual premature degradation of the matrix. The α-Spectrin antibody marks the tracheal cell membranes (red). Scale bars indicate 10 µm (inlay) or 100 µm. (**C**) Confocal images of ultrathin epidermal sections of larval immunostainings. The wt epidermis (upper image) contains a tight cuticle (indicated by the yellow line) that shapes the straight appearance of the epidermis (arrow) of the animal. In *obst-A* mutants (lower image), the epidermal chitin matrix (yellow line) lost its stability and integrity, resulting in a fragile, wrinkled epidermis (arrow) with partial lesions between cells and cuticle. The Obst-A and Knickkopf antibodies detect the chitin matrix. The Alexa633 conjugated wheat germ agglutinin (WGA, in blue) is a lectin that marks chitin and membranes. Scale bar indicates 10 µm. (**D**) Transmission electron microscopy images reveal the soft non-lamellar (upper image, arrows), tracheal and tight lamellar (lower image, arrows point to chitin-lamellae) epidermal chitin matrices. Images were provided by Dr. Dietmar Riedel (Electron Microscopy Group, Max-Planck-Institute for Biophysical Chemistry). Note the fine twisted plywood-like structure of lamellar chitin-protein planes in the epidermis. Red lines indicate epidermal cell surfaces. Scale bar indicates 0.5 µm.

**Figure 3 materials-15-01041-f003:**
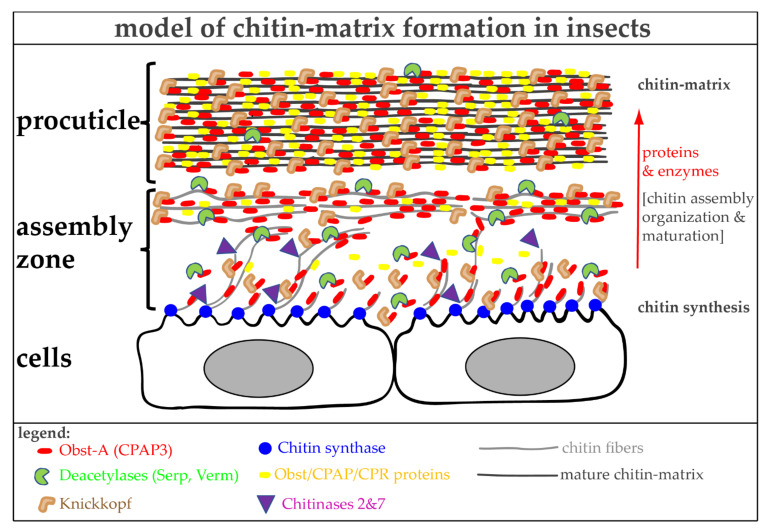
Genetic control of chitin matrix formation in *Drosophila melanogaster*. The chitin-binding protein Obst-A (CPAP3-A) operates as a hub for chitin matrix formation. Within the assembly zone, Obst-A recruits chitin to a scaffold where it places chitin deacetylases (Serp and Verm) and Knickkopf for chitin matrix assembly and maturation into nanofibrils and the subsequent accumulation into compact fibers of the chitinous procuticle. Obst/CPAP family members act synergistically in this process. The genes and their discussed products, the hub of chitin matrix formation, are evolutionarily conserved among arthropods, indicating their general necessity for cuticle biology.

**Figure 4 materials-15-01041-f004:**
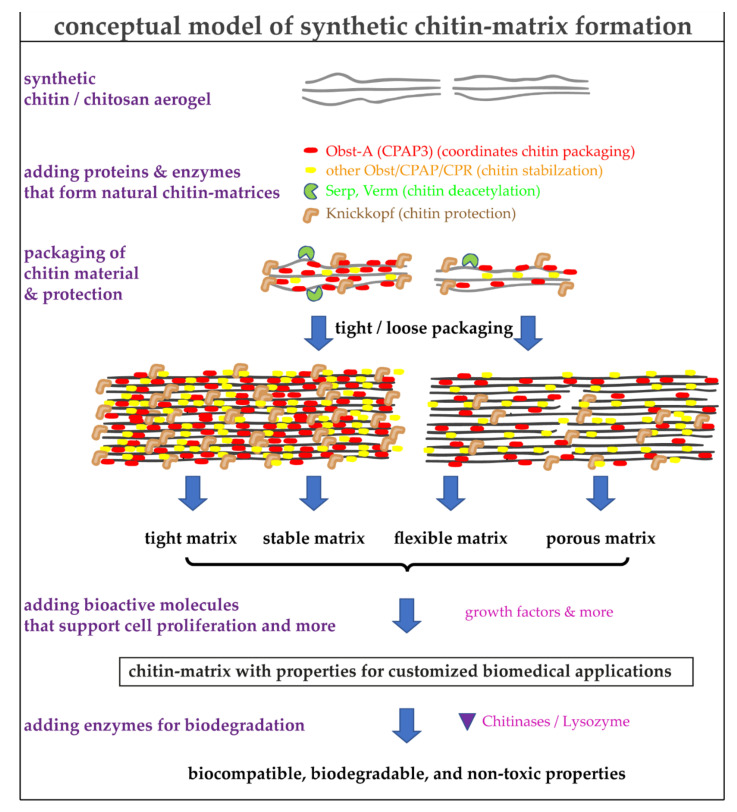
A conceptual model for the usage of critical proteins that control properties of synthetic chitin matrices. The known functions of the proteins are indicated. The combination of synthetic chitin-based aerogels with proteins controlling the chitin packaging and additional bioactive molecules (discussed in the text) will significantly improve and adjust the properties of the materials for customized biomedical applications. The eco-friendly biodegradability is an additional advantage of these new materials.

**Table 1 materials-15-01041-t001:** Advantages and disadvantage of wet-gel network formation methods.

Matrix Formation	Bonding	Advantage	Disadvantage
Chemical crosslinking	Covalent bonding	Improving network strength results in rigid matrix formation providing high mechanical strength	May become brittle
Thermal energy	Van der Waals force	Reversible in wet-gel matrix formation	Gel transition temperature may be very high (>40 °C) or it is controlled by additives
Ionic crosslinking	Van der Waals force	Strong interaction with biological medium; advantage of using the ionic matrix as carrier; buffering at biological medium	High volume shrinkage while supercritically dried aerogel formation
Reversing pH of medium	Van der Waals force	Most employed economic path and easy to handle; possible to functionalize with sensitive biological molecules in neutralized post wet-gel matrix	Inappropriate for functionalizing the chitin/chitosan molecules with sensitive biological molecules
Non-solvent induced phase separation	Van der Waals force	Macropore channels are formed	Non-aqueous medium was used as non-solvent
Percolation	Van der Waals force	Highly crystalline wet-gel matrix is generated	Precursors should be produced in situ; should be stored or transported with high care and caution; not forming wet-gel matrix

**Table 2 materials-15-01041-t002:** Examples for non-protein/peptide-based chitin/chitosan aerogel matrices, reporting good biocompatibility after in vitro tests and high water absorption capacity.

Aerogels	Crosslinked/Blended with Additives	Observed in Biomedicine Application
Chitosan	Crosslinked with diatom-biosilica by in situ polymerization of dopamine (Michael-type cycloaddition) [137]	Improved hemostatic performance.
	No additives [143]	Improved hemostatic performance
	Blended with alginate producing polyelectrolyte complex [144]	Efficient antibacterial activity (*Staphylococcus aureus* and *Klebsiella pneumoniae*) and effective wound closure
	Crosslinked with itaconic acid using epichlorohydrin [145]	Efficient antibacterial activity (*Corynebacterium* *glutamicum* and *Escherichia coli*)
	Vancomycin, drug-loaded chitosan aerogel [97]	Efficient antibacterial activity (*Staphylococcus aureus*) and drug release kinetics
	Cellulose nanofibers dry-crosslinked in the chitosan matrix [146]	Shape recoverable foam material under wet conditions showing improved hemostatic performance
Chitin	No additives, electrophoretic deposition of chitin nanoparticles [138]	Accelerate wound healing and reduce scar area in comparison with cryogels

## Data Availability

Not applicable.

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
