# Peer review of "Improving Polysaccharide-Based Chitin/Chitosan-Aerogel Materials by Learning from Genetics and Molecular Biology"

_materials, 2022, doi:10.3390/ma15031041_

Round 1

Reviewer 1 Report

The uploaded version of the manuscript has a poor layout, since it is a version with the comments and track changes of the authors, not suitable for the reviewers to check. 

Figures are not suitable for publication in quality or display. 

English must be improved. 

Author Response

1. The uploaded version of the manuscript has a poor layout, since it is a version with the comments and track changes of the authors, not suitable for the reviewers to check. 

We a very sorry for this. We submitted a blank ready to read version and, for better convenience, an additional second version showing all changes marked in red. The author track changes were not activated in our submitted versions.

2. Figures are not suitable for publication in quality or display. 

Thanks for this general but unfortenuately less specified comment. We believe that the Reviewer is a recognized expert, and we take his concerns very seriously, but we wonder very much which Figure is not suitable and why. Data Images presented in Fig 2 are adapted in quality and display from original articles already published in well-established professional journals (see Ganesan et al., Chemistry, 2018; Pesch et al., JBC, 2015; Behr & Riedel, Sci Rep, 2020). Similarly, Models in Figs 1 and 3 are adapted in quality and display to already published models about wet-gel-network formation, and chitin-matrix assembly and maturation (see Chaudhari et al., PNAS, 2011; Pesch et al., JBC, 2015; Pesch et al., Sci Rep, 2016; Liu et al., J Insect Physiol. 2019). Figure 4 provides a graphical summary of our new concept model and meets quality standards concerning production of images.

Therefore, we disagree with the Reviewer's comment for the above reasons and firmly believe that they meet the quality and display guidelines of the Journal.

3. English must be improved. 

Thanks, for your input. Due to additional proof reading by international colleagues (Canada) we were able to improve it.

Reviewer 2 Report

This research is under the scope of this journal; the topic is interesting for readers and this research deals with potentially significant knowledge to the field and an open new way for future studies.

The authors improved the quality of the manuscript after the reviewer's indications.

Author Response

This research is under the scope of this Journal; the topic is interesting for readers and this research deals with potentially significant knowledge to the field and an open new way for future studies.

The authors improved the quality of the manuscript after the Reviewer's indications.

We thank the Reviewer for the very constructive and helpful criticism which improved our manuscript

Reviewer 3 Report

Comment:

In this manuscript, Behr and Ganesan systematically reviewed current advances in synthetic chitin-chitosan matrices for biomedical applications. Various mechanisms of synthesizing chitin matrices were reviewed and explained. Interestingly, the authors covered protein-based and genetic approaches that could control the formation of natural chitin matrices. These directions may inspire more efforts in applying chitin-based biomaterials for translational use. While this is well-structured review, there are still some questions that need to be addressed (see comments below). These issues necessitate a minor revision to this manuscript before it can be considered for acceptance.

Additional Comments:

  1. It would be informative to include an “advantage/disadvantage” table for the variation gelation approaches in Figure 1. This will clearly deliver the comparison among these interesting methods.
  2. The discussion of protein-controlled chitin formation is interesting. Has there been any examples of controlling chitin/chitosan aerogels through specific proteins or genetic cues? If so, please highlight these examples.
  3. In Figure 4, it would be informative to list the names of a few key proteins that form natural chitin-matrix. If possible, please also include how they might affect the chitin-matrix properties.

Author Response

First of all, we thank the Reviewer for efforts, kind words, and constructive suggestions to improve our manuscript. Please, find our point-by-point answers to your questions as follows:

Additional Comments:

1. It would be informative to include an "advantage/disadvantage" table for the variation gelation approaches in Figure 1. This will clearly deliver the comparison among these interesting methods.

We added the table 1 showing "advantage/disadvantage of different gelation approaches in the manuscript.

2. The discussion of protein-controlled chitin formation is interesting. Has there been any examples of controlling chitin/chitosan aerogels through specific proteins or genetic cues? If so, please highlight these examples.

This is a great point. Many recent examples show that chitosan aerogels are useful to entrap, reinforce, or even interact with proteins, drugs, or bioactive material, making the aerogels a fascinating biomedical product. Therefore, the potential of the interaction of chitin controlling proteins that modulate aerogels is evident and is one essential thesis of our final concept model. We highlight this as the final paragraph of chapter 7. Nevertheless, this is a new interdisciplinary concept that combines knowledge of chemistry and biology, but to our knowledge, there is no current example showing this in more experimental details.

3. In Figure 4, it would be informative to list the names of a few key proteins that form natural chitin-matrix. If possible, please also include how they might affect the chitin-matrix properties.

Thanks for this fantastic comment. The idea to include critical proteins and how they affect chitin-matrix formation is extremely useful. We revised Figure 4 and firmly believe that the idea strongly improves our final conceptual model.

This manuscript is a resubmission of an earlier submission. The following is a list of the peer review reports and author responses from that submission.

Round 1

Reviewer 1 Report

The review paper entitled "Engineering perfect aerogel materials using genetics and molecular biology for future biomedical usage" by Behr and Ganesan, was submitted for its consideration to be published in MDPI Materials. 

The topic is interesting to be considered and developed as a review type article. Nevertheless, the presentation, data and information presented in the current version of the paper is not suitable for publication. I advise its rejection. 

English style must be changed in order to be suitable for a scientific publication. Abstract should be re-written since you cannot start with a question, nor repeating this rhetorical style with further questions in the same abstract. 

A review paper should include sections regarding the content, not the usual sections of methods, results and discussion. 

Figure 1 does not provide further useful information. 

Figures are not consistent among them (format, font, layout)

Most of the length of the paper is due to references, which have an unusual format in order to extend the length. 

Author Response

First of all, we would like to thank all reviewers for their efforts and time in reviewing our article. In addition, we appreciated all criticism and comments to improve our manuscript. We think that this is indeed the case for this revised version. In the following, we address all comments, respond to them step by step, and implement them in the manuscript.

Reviewer 1

The review paper entitled "Engineering perfect aerogel materials using genetics and molecular biology for future biomedical usage" by Behr and Ganesan, was submitted for its consideration to be published in MDPI Materials. 

1.) The topic is interesting to be considered and developed as a review type article. Nevertheless, the presentation, data and information presented in the current version of the paper is not suitable for publication. I advise its rejection. 

We thank the reviewer for the comment that our topic is very interesting and appears to be suitable for a review article of this journal. Therefore, we worked hard to implement all requested comments and requirements of the reviewer and hope for a more positive view.

In this context, we would like to emphasize that this review intends to focus on new chemical and genetic tools with the idea that both could be combined in the nearer future. There are many other ways to synthesize very promising chitin/chitosan-based matrices, and it is even possible to receive natural chitin matrices, although with limited access. We have clarified all these points in this revised version.  Furthermore, we substantially extended descriptions of potential biomedical usage, we discussed limitations and further modifications of chitin/chitosan-protein-based materials. For a better overview, we will submit the revised manuscript and a version in which all changes are marked in red. Furthermore, this revised version now appears as a well-balanced review manuscript concerning the biomedical, chemical and biological aspects. We think that we have achieved a substantial and manifold improvement of our manuscript contents.

2.) English style must be changed in order to be suitable for a scientific publication. Abstract should be re-written since you cannot start with a question, nor repeating this rhetorical style with further questions in the same abstract. 

We are thankful for this critical comment and apologize for the unusual style. We have substantially corrected the English style of our manuscript. As suggested by the reviewer, we deleted questions and rhetorical style, which included the first four sentences of the abstract and all other parts of our manuscript. This is apparent when looking to all changes. We also had proofreading of the manuscript by colleagues from Canada, and English teachers. Therefore, we do not list the numerous changes here but offer a manuscript version in which all changes are marked in red. We think that the English style improved in the revised version and, above all, is hopefully acceptable for the reviewer.

3.) A review paper should include sections regarding the content, not the usual sections of methods, results and discussion. 

We fully agree with the reviewer, and we are pleased with this suggestion. Our original manuscript style did not contain these sections, but we had to submit to the journal's style, which asked for these unusual sections. We initially followed the journal policy: "Structured reviews and meta-analyses should use the same structure as research articles and ensure they conform to the PRISMA guidelines." Independently from this, we divided the sections in our first submission using headings regarding the content. Nevertheless, we discussed your issue with the editor, and it turned out that it is not necessary to follow this unusual structure. We, therefore, decided to follow your suggestions and divided the review manuscript into the five main sections regarding the contents (1. Chitin/Chitosan matrices and their biomedical potential; 2. Aerogel based chitin/chitosan matrices; 3. Insect chitin matrices and the underlying genetics 4. Advantages and limitations of chitin/chitosan-protein materials 5. Conclusion). We hope that this is now much more convincing.

4.) Figure 1 does not provide further useful information. 

We agree with the reviewer. Our initial idea was to provide help for readers not familiar with chitin and chitosan. Nevertheless, we eliminated figure 1, as suggested by the reviewer, since it can be found in numerous textbooks.

5.) Figures are not consistent among them (format, font, layout)

We are very sorry for this point. As suggested by the reviewer, we unified figures and used the "Palatino Linotype" font used by the journal and similar font sizes for headings and descriptions. New Figures 1 and 3 are both graphical summaries about synthetic and natural chitin-matrix generation, respectively. However, new Figure 2 contains a different layout since it shows images of the products of synthetic aerogel and insect chitin-matrices and, in the case of insects, the genetic function of one essential gene required for chitin-matrix production. For Figure 2 we used Palatino Linotype" font and size for headings and descriptions comparable with Figures. 1 and 3. Nevertheless, the size differences of the Figures are due to the Word mask format of the magazine. Therefore font size may appear different even though it is the same in our original files. However, the sizes of figures usually change again during the final editing.

6.) Most of the length of the paper is due to references, which have an unusual format in order to extend the length. 

We are very sorry for this. When copying our Citavi reference list from word.doc into the predefined journal mask, font size and style appeared immensely increased, and overall size was not changeable. Even decreasing the font size to eight did not change the overall appearance. At no time were we interested in artificially extending the length of our article, especially since the references are numbered. In the revised manuscript, we finally used the journals "citation" style for our references, which unified the references' font, style, and size.

Reviewer 2 Report

Dear Authors,

Manuscript was properly written and a lot of aspects were included.  Do you know, if there is any possibility to obtain chitin/chitosan aerogel matrix using biotechnology (bacteria, fungi)? This process could be similar to engineering process described in manuscript?

Manuscript was written in a concise manner and the main reference point was the nature. Authors concentrated on the polysaccharide-based chitin/chitosan matrix, that is why the title should be clarified (it is too general, others kind of matrices were not presented). There are some parts of the manuscript, which are too broad (Authors mainly focused on the natural and biological aspects). The value of this work could be increased by adding more informations how it could be use in medicine, why is so important for improving human health.

Author Response

First of all, we would like to thank all reviewers for their efforts and time in reviewing our article. In addition, we appreciated all criticism and comments to improve our manuscript. We think that this is indeed the case for this revised version. In the following, we address all comments, respond to them step by step, and implement them in the manuscript.

1.) Dear Authors, Manuscript was properly written and a lot of aspects were included.  Do you know, if there is any possibility to obtain chitin/chitosan aerogel matrix using biotechnology (bacteria, fungi)? This process could be similar to engineering process described in manuscript?

We thank the reviewer for the positive comment on our manuscript.

In nature, chitin is the second-most abundant natural biopolymer after cellulose and can be extracted from some sources, such as crabs, lobsters, shellfish, and fungi, but access and amount are often limited. Therefore, we included this statement (lines 38-41) and reference Number 5, which discussed the biomedical usage of such natural sources.

Cellulose-based porous materials are known from Bacteria. It is reported to be a slow process. But we have not got attention of any publication reporting porous materials of chitin or chitosan using biotechnology. Non-animal based chitin and chitosan are sold in commercial markets for the production of food-grade products. So it can be interesting to produce porous materials directly using biotechnology for the biological application as it will be a non-animal based material.

2.) manuscript was written in a concise manner and the main reference point was the nature. Authors concentrated on the polysaccharide-based chitin/chitosan matrix, that is why the title should be clarified (it is too general, others kind of matrices were not presented).

We agree with the reviewer and are thankful for preciseness and criticism. To clarify the title, we added "chitin/chitosan-"to the title.

3.) There are some parts of the manuscript, which are too broad (Authors mainly focused on the natural and biological aspects). The value of this work could be increased by adding more informations how it could be use in medicine, why is so important for improving human health.

We thank the reviewer for this suggestion and added the requested important information how chitin/chitosan could be used in medicine and why. We added the information about biomedical use and its importance in the new section 1 that we named "Chitin/Chitosan matrices and their potential for biomedical usage". Due to increasing this biomedical point of view, the review manuscript appears more balanced between medical/chemical/biological disciplines. We think that our changes and substantial additions significantly improved the manuscript and hope it will convince the reviewer.

Reviewer 3 Report

This research is under the scope of this journal; the topic is relevant for readers, and this research deals with potentially significant knowledge of the field. 

 However, there are some concerns about the present manuscript:

Abstract 

  • The use of personal pronouns should be avoided. Example “We compared”.

Introduction 

  • Page 1 Lines 33.34 “Finally, chitosan possesses antimicrobial, antioxidant, anticarcinogenic, and anti-inflammatory properties” And also very good biocompatibility, for support this sentence, Please, read this article, Palma et al. (DOI: 10.1016/j.joen.2017.03.005) was studying the Lyophilized gel chitosan (CS) scaffold in the dentistry application, it was investigated in an animal study the usage of chitosan scaffolds with very good biocompatibility for dental pulp regeneration (Regenerative Dentistry). CS was added inside the root canal dentine walls to see recovery of dental tissues.
  • In the final introduction, explain how this narrative review was divided into subtitles!

Results

-  Please, clarified more the limitations for the use of these new materials? 
- And also clarified the future perspectives in the discussion. 

Conclusion
-
Need to add short conclusions of this study after discussion.

References

  • But references are not standardized. The titles of references have a different format, 
    the title of the article is written in capital letters at the beginning of words, others only in lower case. Also, the standardized format of presentation in the journal's name. Because names have been written in a different format, one is not abbreviated, others are not.

Author Response

First of all, we would like to thank all reviewers for their efforts and time in reviewing our article. In addition, we appreciated all criticism and comments to improve our manuscript. We think that this is indeed the case for this revised version. In the following, we address all comments, respond to them step by step, and implement them in the manuscript.

This research is under the scope of this journal; the topic is relevant for readers, and this research deals with potentially significant knowledge of the field. 

 However, there are some concerns about the present manuscript:

1.) Abstract 

  • The use of personal pronouns should be avoided. Example "We compared".

We thank the reviewer for suggestions and deleted personal pronouns from the abstract. We have rewritten the abstract to match the generally accepted scientific style.

2.) Introduction 

  • Page 1 Lines 33.34 "Finally, chitosan possesses antimicrobial, antioxidant, anticarcinogenic, and anti-inflammatory properties" And also very good biocompatibility, for support this sentence, Please, read this article, Palma et al. (DOI: 10.1016/j.joen.2017.03.005 ) was studying the Lyophilized gel chitosan (CS) scaffold in the dentistry application, it was investigated in an animal study the usage of chitosan scaffolds with very good biocompatibility for dental pulp regeneration (Regenerative Dentistry). CS was added inside the root canal dentine walls to see recovery of dental tissues.

We are thankful to the reviewer for this important information. We included this in our new section, "Chitin/Chitosan matrices and their potential for biomedical usage," see lines 92-93, and added the reference.

  • In the final introduction, explain how this narrative review was divided into subtitles!

We want to thank the reviewer for this important information. Because reviewer one suggested changing our initial sections, which are unusual for review articles (Introduction et.), we now divide the manuscript due to its content into five sections: 1. Chitin/Chitosan matrices and their biomedical potential; 2. Aerogel based chitin/chitosan matrices; 3. Insect chitin matrices and the underlying genetics 4. Advantages and limitations of chitin/chitosan-protein materials 5. Conclusion.

Nevertheless, the first section describes the biomedical usage and advantages in more depth, and its final paragraph the further division into subtitles of chemical and biological contents (lines 119 – 128):

“There are many promising examples to produce synthetic chitin/chitosan-based matrices, such as chitosan hydrogels, films, porous scaffolds, textile fibers, and others, and to optimize their functionality and bioactivity as biomedical products [19,22,23,27–29]. Among them, the aerogel technique provides powerful tools for technical modifications, such as blending, crosslinking, and modifying the microstructural and physical properties to produce chitin/chitosan matrices (see section 2). In addition, the numerous variations of chitin-matrix architecture in insects suggest a bioinspired view to the natural molecular toolbox, namely the genes encoding for the proteins and enzymes that organize insect chitin-matrix formation and architecture of chitinous cuticles (see section 3).”  

We hope that is convincing for the reviewer.

3.) Results

-  Please, clarified more the limitations for the use of these new materials? 

-  And also clarified the future perspectives in the discussion. 

We extended the description about biomedical usage and future perspectives (Section 1), and added a new section (4. Advantages and limitations of chitin/chitosan-protein materials), to discusses the challenges, limitations, and future perspectives when designing and producing these new materials.  

4.) Conclusion
-
Need to add short conclusions of this study after discussion.

We added a short conclusion (lines 385 – 398), resulting in a model shown in figure 4.

5.) References

  • But references are not standardized. The titles of references have a different format, 
    the title of the article is written in capital letters at the beginning of words, others only in lower case. Also, the standardized format of presentation in the journal's name. Because names have been written in a different format, one is not abbreviated, others are not.

We are sorry for this, which was in most cases a handling problem with the Citavi reference manager. We unified the appearance of the references in this revised version to the Journal style and checked; this includes well-known abbreviations of journals presented in PubMed and the beginning of titles with capital letters. 

Round 2

Reviewer 1 Report

Changes carried out are not enough to make this paper suitable for publications. It does not follow the requirements for a review paper in terms of extension, number of references (and time period), figure representation, content... 

I do not recommend its publication. 

Author Response

Dear Editors,

Dear Reviewers,

Thanks for the opportunity to submit a revised version of our manuscript.

We are pleased that Referee 2 and Referee 3 were fully satisfied with our revised manuscript and recommend its publication. We are very grateful for this and appreciate their comments and helps during the revision process.

However, Referee 1 was not fully satisfied.

We used this revision as an additional incentive to improve our manuscript along the line of argumentation of Referee 1. First of all, we clarify in the manuscript, and here again, that we do not present a formal review about chitin/chitosan materials used in the past. A search in PubMed shows 537 entries for review articles about chitin/chitosan materials, altogether 72 alone in 2021! To write another one would present nothing new. In other words, our manuscript is not a review paper describing all kinds of known chitin-materials along “time periods”.

We aim to present and focus on a new method, idea, and concept on how chitin/chitosan-materials should be improved by natural biomimetic aspects, meaning the use of critical proteins and enzymes that organize the diversity of natural chitin-matrices in the animal kingdom. Among hundreds of identified proteins, we provide the critical knowledge to learn that only a handful of critical proteins/enzymes are enough to adjust chitin/chitosan properties. We describe the molecular mechanisms of how these key proteins and enzymes modify the diversity of chitin-matrices and, therefore, could be a valuable tool for generating chitin-materials. We further clarify why the aerogel technique could be a breakthrough for adding crucial proteins/enzymes to combine biology and chemistry in a new bionic-inspired way.

When searching PubMed for this combination of biologically inspired engineering, we did not find any entry. This shows that our manuscript represents a new concept. The reasons for this are evident, as the essential proteins and enzymes have only been identified in recent years. Additionally, the research on aerogels of chitin/chitosan has been rapidly growing in the last years, providing a powerful technique for bionic research. 

Nevertheless, we took the criticism of Referee 1 seriously.

-First, we checked and improved the citations. In the current manuscript is no content left without citing the corresponding reference. Our manuscript now cites 232 references, a good standard among the chitin/chitosan review articles.

-Second, we extended the content concerning natural chitin, biomedical usage of chitin-materials on wounds, and chitin-aerogel fabrication. We even present a list of the newest class of porous aerogel materials used for wound healing. Altogether, the review is now very comprehensive without blurring the key message. All changes are listed below.

-Third, Figure 1 has been extended by illustrating ionic bonding. The subjective perception of an image is one thing. The objective design of an image is another matter. We authors have generated dozens of scientific illustrations and published them in a wide variety of high-ranking journals. Thus, the production of scientific illustration is nothing new for us. The figures in this manuscript were generated with Adobe products and designed in a high standard scientific way comparable with other review articles in high-impact journals, indicating that the representation of the figures is of a high standard. This high scientific standard also includes comprehensive figure legend descriptions and the main text.

It further includes labeling of all key items. To sum up, the four figures of our manuscript include all essential information’s to understand the manuscript and the new biomimetic concept at a glance which meets tutorial guidelines for scientific figures and images (see list of figure characteristics below).

The manuscript has grown to include numerous topics and references while still focusing on the new bionic chitin/chitosan material concept.

We are looking forward to hearing from you.

With kind regards

Matthias Behr , Kathirvel Ganesan

The figures and schematic drawings of this manuscript meet the official tutorial guidelines for figures and images (see Springer guideline, online)

https://www.springer.com/gp/authors-editors/authorandreviewertutorials/writing-a-journal-manuscript/figures-and-tables/10285530

Figures

Figures are ideal for presenting:

Images/Data plots/Maps/Schematics

Just like tables all figures need to have a clear and concise legend caption to accompany them.

Images:

Include scale bars

Consider labeling important items

Indicate the meaning of different colours and symbols used

Schematics

Schematics help identify the key parts to a system or process. They should highlight only the key elements because adding unimportant items may clutter the image. A schematic only includes the drawings the author chooses, offering a degree of flexibility not offered by images. Label key items. Provide complementary explanations in the caption and main text

List of changes:

Abstract

Line 16: deleted: “This review discusses the generation of synthetic chitin/chitosan-matrices with a bioinspired view adapted from the natural molecular toolbox.”

Added: “Synthetic chitin/chitosan hydrogel and aerogel techniques provide the advantages for improvement with a bioinspired view adapted from the natural molecular toolbox.”

  1. Chitin/Chitosan-matrices and their potential for biomedical usage

Line 32, expanded sentence: The reasons for this are manifold since natural and synthetic chitin/chitosan-matrices can be flexible and, at the same time, very robust and scratch-resistant

Line 39, added “non-toxic” and “Chitin”

Line  41, added “since many sources are part of a global food market”

Line 53, added “Chitosan biodegradation can be triggered by enzymes such as lysozyme (muramidase) [22–25]. Lysozyme is expressed in the normal human intestine and other mucosal epithelial cells [26–28]. Moreover,..”

Line 59, added: “Collectively, chitosan and its composites are promising matrices for active pharmaceutical and other bioactive compounds. The chitosan gels show bioavailability for drugs, enzymes, and proteins recently summarized by Takeshita and colleagues [32].”

Line 100, added: “Further, chitin films showed no allergenicity upon dressing wounds of rats [58]. Yet, there is no apparent effect of chitosan in terms of allergy or severe inflammation, as summarized by Muzzarelli [7]. “

Line 110, added: “Moreover, chitin-gauze and -spray revealed satisfactory healing of a variety of traumatic wounds [60]. Finally,..”

Line 114, added: “The numerous synthetic chitosan-based scaffolds used to promote wound healing are summarized recently by Merzenderofer [5]. However, wound healing comprises many aspects, namely cleansing, granulation/vascularization, and epithelialization. These depend on an optimal microenvironment and cytotoxic factors [62,63].”

Line 120, added: “…we focus on a new concept combining…”

Line 125, added: “Among the many matrices, aerogels are porous materials with unique properties such as high porosity, low bulk density, high surface area, ultra-light, and high water uptake [2]. The promising use of natural-based and synthetic aerogels for fluid management, healing, and regeneration of wounds and the increasing trend of articles published per year is summarized recently by Bernardes and colleagues [2,67,68]. Such chitin/chitosan-aerogels show advanced properties such as inhibiting the Staphylococcus aureus growth, a widespread cutaneous pathogen responsible for the great majority of bacterial skin infections in humans [69,70]. Moreover,…”

Line 135, added: “Hou and colleagues state in their recent opinion that for the future’s advanced technologies, using chitin will ultimately drive many innovations and alternatives using biomimicry in materials science [4]. Natural…”

Line 137, moved the paragraph introducing “Natural chitin-matrices…..” to the end of this section 1.

Line 143, added: “…due to coherent scattering in the periodic arrays of multilayered chitin/chitosan matrix structures [4]. The wide range of uses becomes even clearer considering that the chitin-matrix is formed by numerous organisms, such as fungi, worms, and arthropods, including crustacea and insects [76].”

  1. Aerogel based chitin/chitosan matrices

Line 220, added:” The covalent crosslinking of carboxymethyl chitosan with collagen peptides using water-soluble carbodiimide (1-(3-dimethylaminopropyl)-3-ethylcarbodiimide hydrochloride) and N-hyydroxysuccinimide was recently shown [101]. Further purification..”

Line 227, added:” Morphology and topography of the chitin/chitosan matrices receive close attention in designing the wound dressing material. While ambient drying or vacuum drying the wet-gel matrices of chitin/chitosan materials, the existing surface tension between solid, liquid, and gas can pull the nanostructures together. At the same time, the fluid becomes gas, which destroys the natural structure [102]. Therefore, it results in more than 90% of volume shrinkage. These materials are named xerogels [103]. The complete collapse of pores can be confirmed by nitrogen adsorption-desorption analyses, mostly showing no considerable values if the solvent medium in the matrix is water or ethanol [103,104]. The methods for preserving pores employed in the laboratories are called freeze-drying (sublimation of frozen water to gas) and supercritical drying (solvent exchange of solvent with supercritical gas [102,104]. They fabricate highly porous materials as the fluid in the wet-gel nanostructures of chitin/chitosan can be replaced with air [32].

In the freeze-drying technique, the natural gel nanostructures are destroyed due to Vander Waal’s interactions promoting significant aggregation of nanofibers. As a result, the specific surface area becomes low (70 -110 m2/g) [104]. However, due to the frozen water molecules (ice crystals), the pores generated by sublimation can display the material to be highly porous. These foam-like materials are named cryogels [102,104]. In some reports, it was demonstrated to use tert-butyl alcohol instead of water in the gel matrix to avoid structural damage [105]. Due to the low surface tension and high freezing point of tert-butyl alcohol compared to water, freeze-drying reduces the capillary forces between solvent and gel matrix and maintains the three-dimensional network structure [105].

Line 250, added: “ In supercritical drying, environment-friendly supercritical gas such as carbon dioxide is employed. In this drying process, the fluid in the gel matrix is switched to a suitable solvent such as simple alcohol (e.g., ethanol), which has good miscibility with CO2. Then the alcohol in the gel matrix is saturated with CO2 under supercritical conditions (>31 °C and >73.8 bar) in an autoclave [102].

Finally, supercritical carbon dioxide gas is degassed at above 40°C, and the matrix is filled with air [102]. Approximately…”

Line 269, added:” There are huge morphological differences between ambient dried, freeze-dried, and supercritically dried materials [104]. Xerogels, cryogels, and aerogels do not show similar interactive properties with the wound surface. For instance, Guo et. al. demonstrated in the recent report that the morphology of aerogels of chitin promoted the excellent healing process, accelerated macrophage migration, enhanced fibroblast proliferation, induced collagen deposition, and promoted granulation and vascularization in comparison with cryogels. Moreover, after 11 days of the wound healing process, the scar area appeared reduced and was significantly smaller in the case of aerogels than cryogels. [106].

In the last decades, the dry membrane, scaffolds, fibers, and hydrogels of chitin/chitosan in wound healing applications have been explored, and mechanisms have been analyzed in vitro and in vivo [33,55,107–110]. Despite that, the finely distributed nanostructures and open porous network of aerogels of chitin and chitosan will be a new class of porous materials for this field of application (examples are listed in Table 1). The aerogels of chitin/chitosan-functionalized with protein/peptides are promising candidates for wound healing applications. The materials possess specific surface area, high sorption capacity, biocompatibility, antimicrobial property, interactive surface functionalities, and the structural similarity of the skin’s extracellular matrix [2,106].

  1. Insect chitin-matrices and underlying genetics

Line 289, added:” Chitin-based matrices are the most prominent part of the arthropod cuticles”

Line 296, added:” it must be coordinated clock-wise throughout development”

Line 308, added:” Such linear chitin polysaccharides occur naturally in three crystalline allomorphs known as α-, β-, and γ-chitin with different orientations of the microfibrils [9,140–148]. Chitin sheets are arranged in layers that show antiparallel orientation in α-, and γ-chitin and parallel in β-chitin [149]. This arrangement has consequences on chitin properties; for example, α-chitin has strongest intersheet and intra-sheet hydrogen bonding. In contrast, the β-chitin has weak hydrogen bonding by intrasheets and is characterized by a weak intermolecular force [150,151]. “

Line 326, added: “This first physical inward barrier produces cuticle nano- and macrostructures the surface [163–166]. It is composed of lipophilic compounds including waxes and free cuticular hydrocarbons [167–172]. Therefore,..”

Line 334, added: “…from which it is known that mutations cause harlequin ichthyosis.”

Line 344, added:” The peritrophic matrix surrounding the midgut food is..”

Line 346, added: “Importantly, chitin-chains are embedded in a protein matrix, forming several horizontal lamina layers [115,154,192]. These laminae can appear in several configurations, including helicoidal stacks (a twisted plywood-like structure), which respond to external stress [193]. Those..”

Line 349, assed:”… above described rigid/stiff but also…”

Line 401, added: “Since chitin-deacetylases catalyze the removal of acetyl groups from chitinous substrates, they play critical roles in shaping chitin-matrices of cuticles in Drosophila [188,189,216] and other insects [217–221].”

Figures/Tables

Figure 1 exetnded by ionic bonding

Figure 3 deleted “cell” and “secretion”

Table 1 added

References

Extended list of references to 232 entries

Reviewer 3 Report

The authors modified the article according to the reviewer's instructions. 

Author Response

Dear Reviewers,

We are pleased that Referee 2 and Referee 3 were fully satisfied with our revised manuscript and recommend its publication. We are very grateful for this and appreciate their comments and helps during the revision process.

Kind regards

Matthias Behr & Kathirvel Ganesan